# Quorum Sensing and NF-κB Inhibition of Synthetic Coumaperine Derivatives from *Piper nigrum*

**DOI:** 10.3390/molecules26082293

**Published:** 2021-04-15

**Authors:** Yael Kadosh, Subramani Muthuraman, Karin Yaniv, Yifat Baruch, Jacob Gopas, Ariel Kushmaro, Rajendran Saravana Kumar

**Affiliations:** 1Avram and Stella Goldstein-Goren Department of Biotechnology Engineering, Ben Gurion University of the Negev, Be’er Sheva 8410501, Israel; shlicter@post.bgu.ac.il (Y.K.); kariny@post.bgu.ac.il (K.Y.); yifatbar@post.bgu.ac.il (Y.B.); 2Chemistry Division, School of Advanced Sciences, Vellore Institute of Technology Chennai 600127, India; muthu.ramans2014@vit.ac.in; 3Department of Microbiology, Immunology and Genetics, Faculty of Health Sciences, Ben Gurion University of the Negev, Be’er Sheva 8410501, Israel; jacob@bgu.ac.il; 4Department of Oncology, Soroka University Medical Center, Ben Gurion University of the Negev, Be’er Sheva 8410501, Israel

**Keywords:** quorum sensing, antibacterial, NF-kB, *Piper nigrum*, coumaperine, plant natural-based compounds, amide alkaloids

## Abstract

Bacterial communication, termed Quorum Sensing (QS), is a promising target for virulence attenuation and the treatment of bacterial infections. Infections cause inflammation, a process regulated by a number of cellular factors, including the transcription Nuclear Factor kappa B (NF-κB); this factor is found to be upregulated in many inflammatory diseases, including those induced by bacterial infection. In this study, we tested 32 synthetic derivatives of coumaperine (CP), a known natural compound found in pepper (*Piper nigrum*), for Quorum Sensing Inhibition (QSI) and NF-κB inhibitory activities. Of the compounds tested, seven were found to have high QSI activity, three inhibited bacterial growth and five inhibited NF-κB. In addition, some of the CP compounds were active in more than one test. For example, compounds CP-286, CP-215 and CP-158 were not cytotoxic, inhibited NF-κB activation and QS but did not show antibacterial activity. CP-154 inhibited QS, decreased NF-κB activation and inhibited bacterial growth. Our results indicate that these synthetic molecules may provide a basis for further development of novel therapeutic agents against bacterial infections.

## 1. Introduction

Antibacterial agents are considered to be one of the most important discoveries of the twentieth century. Nevertheless, an increase in the resistance of bacteria to antibacterial agents and a decrease in the development of new antibiotics, according to CDC and WHO reports, are considered a serious worldwide concern [1,2,3]. To date, conventional antibacterial agents have targeted vital features in the bacteria and have led to selective pressure and the emergence of resistance populations [4]. One feature that seems to be under less selective pressure is the bacterial communication system termed Quorum Sensing (QS). This is a process that regulates the gene expression in response to changes in the population density. The bacteria produce and release small molecules known as Auto-Inducers (AI). When the cell density rises, AI molecules reach a threshold concentration and interact with a receptor protein to form a complex [5]. The AI-receptor complex is a transcriptional regulator protein that complex binds to certain DNA sequences and activates target genes. These create additional AI complexes through a positive feedback [6]. It is well-established that QS regulates numerous virulence genes; thus, these genes can be used as possible targets for attenuation and as a treatment for bacterial infection [7,8,9,10,11,12]. The main advantages of such QS inhibitors are the avoidance of rapid selective pressures that conventional antibiotics generate and, thus, may provide a possible treatment for existing infections by multi-drug resistance bacteria. In mammals, bacterial infections alter the immune system and cause an inflammatory reaction that can by itself be an issue.

Such an inflammation reaction is a stress response that includes a number of signals (e.g., cytokines, chemokines, biogenic amines and eicosanoids) that induce changes in diverse biological processes, suppressing the local or systemic homeostasis. The immune responses to infection are tightly regulated inflammation balancing between elimination of the intruder and the possible damage to the tissues [13,14]. Aberrant regulation of the immune responses to an infection may cause an increase in tissue damage, ultimately leading to increased mortality [15]. Therefore, immune modulation may provide an important method of treatment for infection. A promising target for immune modulation is the Nuclear Factor kappa B (NF-κB) transcription factor that regulates over 500 genes involved in inflammation, cell proliferation and apoptosis. The NF-κB family comprises five members: RelA (p65), relB, c-Rel, p105/p50 and p100/p52 [16]. Moreover, NF-κB displays an important role in first-line protection against intruders (e.g., viruses or bacteria) and is evolutionary conserved [17]. Indeed, some pathogens evolved to modulate this transcription system in order to avoid or to take advantage of the immune responses [18,19]. Examples for this is what occurs during long-term infections of *Pseudomonas aeruginosa* in cystic fibrosis patients, as well as what occurs during urinary tract infections. *P. aeruginosa* infections were found to be able to avoid phagocytes and leukocytes and cause tissue damage and apoptosis with rhamnolipids and exotoxins [20]. The QS signal molecule of *P. aeruginosa* 3-oxo-C12-HSL was found to inhibit the degradation of IκB, inhibitor of NF-κB, interestingly, which enables a longer activation of NF-κB, which is likely to promote the persistence of the infection [21]. In an additional example: *Acinetobacter baumannii* pneumonia causes systematic inflammation through the activation of NF-κB, increasing the production of proinflammatory cytokines and increased expression of Toll-Like Receptor 4 (TLR4). Interestingly, treatment with the anti-inflammatory cytokine Interleukin 33 (IL-33) depresses the inflammation storm and decreases the mortality in a model of *A. baumannii*-infected mice [22]. Finding a molecule that can inhibit both QS and NF-κB may therefore present superior therapeutic value against bacterial infections.

In recent years, plant-derived natural products known to contain NF-κB modulatory activity were investigated. Additionally, natural compounds from plants that have long been used to treat microbial infections have gained much attention as a source of quorum sensing inhibitors [23]. Malabaricone C isolated from the bark of *Myristica cinnamomea* inhibited quorum sensing–regulated biofilm formation in *P. aeruginosa* (PAO1) and violacein production by *Chromobacterium violaceum* CV026 [24]. Recently, dietary phytochemicals with medical uses in humans have been investigated for anti-quorum sensing activity. In addition, many plant-derived natural products are known to contain NF-κB modulatory activity, such as curcumin, which inhibits the phosphorylation of IκBα and p65 [16]. The plant-derived curcumin is a major constituent of turmeric that was found to inhibit quorum sensing-regulated biofilm formation in uropathogens [25]. Piperine, a bioactive constituent of black pepper, was also shown to inhibit biofilm formation by interfering with quorum sensing activity in *Streptococcus mutans* [26]. (Figure 1). An additional natural product, Coumaperine, an amide alkaloid found in white pepper (*Piper Nigrum*), structurally similar to Piperine whose bioactivity has also been explored [27]. Recently, our group reported the 5-LOX inhibitory activity [28] and antiproliferative activity of coumaperine derivatives against two cancer cell lines, L428 and A549, and the inhibition of A549 cell migration [29]. However, the anti-QS activity of coumaperine and its derivatives are still unexplored. Therefore, in the present study, we tested 32 derivatives of coumaperine to determine their quorum sensing inhibition (QSI), antibacterial and NF-κB inhibitory activities.

## 2. Results

### 2.1. Synthesis of Monoconjugated Coumaperine Derivatives

The monoconjugated coumaperine derivatives outlined in Figure 2 were synthesized, and different yields were obtained as noted in the figure (47–76%). Detailed synthetic procedures for each individual compound are presented in the Appendix A).

### 2.2. Synthesis of Di- and Triconjugated Coumaperine Derivatives

The di/triconjugated coumaperine derivatives shown in Figure 3 were synthesized, and different yields were obtained, as noted in the figure (12–98%). Detailed synthetic procedures for each individual compound are presented in the Appendix A. The synthesized di- and triconjugated coumaperine derivatives are depicted in Figure 3 and Figure 4. Coumaperine (CP) was acetylated to obtain CP-158 in a moderate (36%) yield. For the present study, piperine was isolated from black pepper [30]. Piperine was deprotected using BBr_3_ to obtain CP-209 in a moderate yield (53%). CP-209 was also synthesized by other methods. *N*-Crotonyl piperidine was condensed with tetrahydropyran-protected 3,4-dihydroxy benzaldehyde in the presence of a base (*t*-BuOK) to obtain tetrahydropyran-protected CP-209. It was then deprotected with TFA to obtain CP-209 in a moderated yield, 50%. Acetylation of CP-209 with acetic anhydride in the presence of a base (Et_3_N) gave the diacetylated product, CP-281-F1 in a moderate yield (41%). CP-262-F1 and CP-262-F2 were synthesized from the corresponding trimethoxy derivative, CP-27 (shown in Figure 3). CP-27 was demethylated using BBr_3_ to obtain monodemthylated derivative, CP-262-F1 and tridemthylated derivative, CP-262-F2 in moderate-to-poor yields (33 and 14%), respectively. The attempt to isolate didemethylated derivatives were not fruitful.

The synthesized compounds were characterized by various spectroscopic techniques, including, FTIR, ^1^H-NMR, ^13^C-NMR, GC-MS and HRMS/elemental analyses. The spectral data matched with structural attributes for each individual compound are presented in the Appendix A. We previously confirmed the structure of CP-155, CP-209 and CP-262-F2 by single-crystal X-ray diffraction [28].

### 2.3. Quorum Sensing Inhibition Assay

The effects of all the synthesized compounds evaluated for quorum sensing inhibition against two bioreporter bacteria strains, *Chromobacterium violaceum* (CV026) and *Agrobacterium tumefaciens* (KYC55) are summarized in Table 1. The monoconjugated derivatives with a simple phenyl group (CP-270), methylenedioxy-substituted (CP-215), and alkyl-substituted compounds (CP-296), were highly active in both tests (Table 1, entry 1–3). The monomethoxy derivative, CP-282 (Table 1, entry 4) was more effective against CV026 than KYC55, while the dihydroxy derivative CP-289 (Table 1, entry 5) was more effective against KYC55 than CV026. The dihydroxy compound CP-237 and cycloalkyl derivative CP-295 (Table 1, entries 6 and 7) displayed moderate activity against both the bacterial systems. The monoconjugated derivative with an electron-withdrawing substituent nitro group (CP-286) exhibited low and moderate activity against CV026 and KYC55, respectively (Table 1, entry 8). CP-273 exhibited low activity and was inactive against CV026 and KYC55, respectively. CP-291 was inactive against CV026 and exhibited low activity against KYC55.

The diconjugated derivative with fluorine substituent CP-154 (Table 1, entry 9) was highly active in both tests. The acetylated coumaperine CP-158 (Table 1, entry10) was more effective against CV026 than KYC55. The diacetylated (CP-281-F1) methoxy (CP-38) and simple phenyl derivatives, CP-9 (Table 1, entries 11–13), displayed moderate and low activity against CV026 and KYC55, respectively. CP-209 showed low activity against both bacterial strains. Coumaperine (CP) CP-32, 147 and 193 exhibited low activity and was inactive against CV026 and KYC55, respectively. CP-123 and 262-F2 were inactive against CV026 and displayed low activity against KYC55. The other diconjugated derivatives, such as CP-10, 27, 50, 184, 194, 209,262-F1 and PIP, were inactive against both the bacteria strains (Appendix A). The triconjugated derivative with the monomethoxy substituent CP-102 (Table 1, entry 14) showed moderate and low QSI against CV026 and KYC55, respectively. The methylenedioxy-substituted derivative (CP-155) and *N,N*-dimethyl derivatives (CP-146) were inactive against both bacterial strains (Appendix A).

### 2.4. Antibacterial Effect

We also evaluated all the molecules that showed anti-quorum sensing activity for the antibacterial properties test against both Gram-positive and Gram-negative bacteria. Interestingly, none of the molecules showed antibacterial action against Gram-negative bacteria, and only three of the coumaperine derivatives: CP-9, CP-154 and CP-147, showed antibacterial activity, exclusively against the Gram-positive bacteria. The antibacterial activities of derivatives showing QSI are outlined in Table 2.

### 2.5. NF-κB Luciferase Reporter Gene Assay

To further explore the possibility of QS molecules as modulators of NF-κB, we also evaluated the effect of the anti-QS active compounds in reducing NF-κB activation luciferase reporter gene NF-κB inhibition. The compounds that showed a reduction in NF-κB activity were further investigated to determine their potency in a dose response experiment and are presented in Figure 5. At these concentrations, the compounds, except for CP-158, efficiently inhibited NF-κB in L428 cells in a dose-dependent manner. To calculate the median inhibitory concentration (IC_50_), the percent of NF-κB inhibition was plotted against the log of concentrations. CP-154, IC_50_ = 58.8 µM, CP-215, IC_50_ = 136.37 µM and CP-286, IC_50_ = 45.42 µM. As a positive control, curcumin was used in two concentrations: 80 and 160 µM. Curcumin did not significantly inhibit at 80 µM (NF-κB activation is 91.16% ± 6.5%, *p* = 0.2). At 160 µM, curcumin significantly inhibited NF-κB (NF-κB activation 19.08% ± 10.7%, *p* = 0.001).

### 2.6. Quantitation of Activated Nuclear p65-NF-κB Fluorescence

To confirm that the derivatives (CP-154, CP-158, CP-286 and CP-215) inhibit NF-κB, as detected by the luciferase reported gene assay, we used immunostaining. A549 cells were treated with different derivatives, CP-154, CP-158, CP-286 and CP-215 followed by the activation of NF-κB with 2.5 ng/mL of Tumor Necrosis Factor α (TNFα) for 15 min. Fluorescent images were taken from 25 fields in each well in duplicate by the Operetta imaging system and analyzed through the Columbus server. Figure 6A,B are visualizations of the differences between inactive p65 (A) and active p65 followed by TNFα incubation (B). To analyze these results, we defined the active population in the Columbus server (Figure 6D) and the relative fluorescence values of the nucleus and cytoplasm (Figure 6C). As shown in Figure 6, CP-154 and CP-215 significantly inhibited NF-κB activated by TNFα, while CP-286 and CP-158 were less active (Figure 6C). The same results were obtained when the percentage of cells with active NF-κB were scored (Figure 6D and Figure 7).

### 2.7. Cell Viability Assay

We tested the cytotoxicity of the compounds that showed QSI on L428 cells as described in Materials and Methods. CP-286, CP-273, CP-296, CP-281-F1, CP-289, CP-270, CP-282 and CP-215 were not cytotoxic at the concentrations tested. CP-9, CP-154 and CP-38 were toxic with an LC_50_ of 24.70 µM, 46.78 µM and 47.44 µM respectively. CP-158 and CP-295 were less cytotoxic, with LC_50_ of 285.38 µM, and 408.27 µM respectively. (Appendix A)

## 3. Discussion

Finding molecules that inhibit QS and NF-κB without affecting bacterial growth and preventing drug resistance is of utmost importance. Agents that interfere with bacterial virulence without promoting -resistant, conceivably, will have important therapeutic applications [23]. In the present study, CP, which resembles piperine in structure and is present in low concentrations (<6 ppm) in white pepper, as well as its synthetic derivatives, were evaluated as quorum sensing inhibitors. Although CP is characterized by important pharmacophores such as the Michael acceptor, phenolic and amide moieties, its bioactivity has not been properly explored, probably due to its low bioavailability and long synthetic protocols. Recently, we reported a simple and a robust methodology for the synthesis of CP and its derivatives [28,29]. Following that synthetic process, CP derivatives depicted in Figure 2, Figure 3 and Figure 4, were synthesized, and characterized for the present study. Initially, the synthesized compounds were evaluated for their QSI properties. These compounds were then examined further for their antibacterial activity, cytotoxicity and NF-κB inhibition activity.

The quorum sensing inhibitory potential of all the CP derivatives was evaluated. CV026 and KYC55 bacteria detect acyl-homoserine-lactone (AHL), a common signal molecule known as autoinducer (AI) of gram-negative bacteria. Different AIs differ in the carbon chain length of the acyl group. CV026 detect *N*-hexanoyl-l-homoserine lactone (HHL) and represent AIs with a shorter carbon chain, whereas KYC55 detect *N*-3-oxooctanoyl-HSL (OOHL) represents AIs with longer carbon chains. Together, these two QS systems are sensitive to a wide range of AIs and serve as sensitive screen models to a wide range of inhibitory molecules [31].

The anti-quorum sensing pattern depicted in Table 1 generally shows that mono conjugated CP derivatives were more effective in comparison to di- and triconjugated derivatives against both bacterial strains, CV026 and KYC55. Thus, shorter conjugation is more effective with most of the derivatives.

Furthermore, monoconjugated derivatives with simple alkyl, phenyl or aryl groups with electron-donating substituents were more effective than monoconjugated derivatives with electron-withdrawing substituents at the aryl group. The diconjugated derivatives with sterically less demanding electron-withdrawing groups, exhibited better anti-quorum sensing activity than the compounds with sterically bulky electron-withdrawing and electron-donating substituents. Similarly, the triconjugated derivatives with sterically less demanding electron-donating substituents were more effective than the derivatives with sterically bulky electron-donating substituents. Thus, monoconjugated derivatives with electron-donating and diconjugated with electron-withdrawing substituents exhibited a high anti-quorum sensing effect against both the bio-reporter bacteria strains, CV026 and KYC55.

The results of the anti-bacterial tests show that CP derivatives that inhibit QS and have the potential to inhibit QS but are not antibiotic, could be used, avoiding the risk of selecting for drug-resistant bacteria. Similar, synthetic derivatives of furanon (a product of marine algae) showed in-vivo QSI activity without antibacterial activity followed by an immune response in the infected animal [32]. Hossain et. al. [33] previously reported that phenolic compounds decrease QS and virulence in PA01 and *Chromobacterium violaceum*. Thus, there is an increasing number of studies, searching for compounds that target non-vital processes to combat bacteria [34].

Interestingly, the compounds CP-215, CP-154, CP-158 and CP-286 inhibit NF-κB in L428 cells and CP-215, CP-154 and CP-158 inhibit NF-κB in both L428 and A549 cell-lines. In principle, the results obtained here confirmed those obtained using the luciferase reporter gene system [35]. CP-286 and CP-158 were less effective in A549 as compared to their effect in L428 cells. The difference may be due to the compounds’ different mechanisms of action, independent or dependent on IκB, since L428 cells lack IkB, while present in A549 cells. Curcumin was used as positive control and significantly inhibit NF-κB activation. The compounds had variable cytotoxicity, CP-215, CP-158 and CP-286 were not cytotoxic. while CP-154 was toxic even at low concentrations.

Thirty-two CP derived molecules were screened for anti QS activity. Twenty three compounds inhibited QS and were further tested for reducing NF-κB activation. Five were found to have both activities (CP-9, CP-154, CP-158, CP215 and CP-286), of which, three were not cytotoxic and did not show antibiotic activity (CP-158, CP-215 and CP-286). These results are summarized in Table 3. CP-9 was previously reported to be cytotoxic and to inhibit NF-κB [29]. Here, we show that CP-9 also has a moderate and low inhibitory effect on QS of CV026 and KYC55, respectively (Table 1) and inhibit bacterial growth of the three gram-positive bacteria were tested (Table 2). These activities suggest that CP-9 and CP-154 may affect multiple targets within bacterial and/or human cells. Interestingly, CP-215, being active but not cytotoxic may show promise as a useful QSI molecule.

The structure of the derivatives mainly differs in the degree of conjugation and/or in the functional group. Some structural features have an impact on the QSI and NF-κB inhibition activities. For example, CP-273 and CP-154 have a similar structure, containing fluoride and differing only in their carbon chain length. Despite their similarity, they perform very differently in their biological activities. CP-273 has low QSI, no anti NF-κB activity (Table 1), and is not cytotoxic (Appendix A). CP-154 on the other hand showed activity but was also cytotoxic (Appendix A). The major differences in their activity might have been determined by the length of their carbon chain. Furthermore, CP-215, PIP and CP-155 differ only in their carbon chain length but show varied biological activity. CP-215 has strong QSI and NF-κB inhibition activities (Table 1, Figure 5, Figure 6, Figure 7 and Figure 8). By contrast, PIP and CP-155 do not inhibit QS or NF-κB. We also compared CP-270, a mono-conjugate derivative and CP-9, a di-conjugate derivative with no functional group on their aromatic ring. While CP-9 had some antibiotic activity CP-270 do not. CP-270 is highly QSI active in both bio-reporters but CP-9 has moderate and low activity in CV026 and KYC55, respectively (Table 1). In this study, CP-270 had no anti-inflammatory activity, while CP-9 demonstrated strong NF-κB inhibition [29].

The unique contribution of the functional group can be observed in the couple CP-38 and CP-147, both di-conjugates, where the differences in their structure area thiomethyl functional group (–SCH_3_) and in CP-38 a methoxy (–OCH_3_) moiety. This relatively small change has a significant impact on their functions, CP-147 inhibits *Staphylococcus aureus* growth (Table 2), has a low QS inhibition effect on CV026, does not affect QS in KYC55 (Table 1) and has no effect on NF-κB. The activities of CP-38 were listed above (and summarized in Figure 8). The different functional group among them is interesting, where oxygen is more active than thio at this position.

Here, we show that synthetic derivatives of natural compounds often have improved therapeutic values over the parental molecule, as they enable us to solve problems of solubility, toxicity, activity and specificity. In conclusion, QS inhibition is a promising approach to prevent bacterial virulence, focused on the disturbance of bacterial communication. Coupled with the ability to inhibit NF-κB, we believe of our derivatives are suitable candidates for further development as therapeutic substances to prevent complications resulting from bacterial infections.

## 4. Materials and Methods

### 4.1. Chemistry

#### 4.1.1. Materials

All the commercially obtained reagents/solvents for the synthesis of coumaperine and its derivatives were purchased from Spectrochem^®^ (76/1, Industrial Suburb, Behind Mysore Sandal Soap Factory, Yeswanthpur, Bangalore-560022, India.), SRL^®^ (608-B, Satellite Gazebo, Andheri Ghatkopar Link Road, Chakala, Andheri (E), Mumbai-400 099, India.), Alfa Aesar^®^ (403-404, Delphi ’B’ Wing, Hiranandani Business Park, Powai, Mumbai-400076, India.), RANKEM^®^ (No. 139, BDA Industrial Suburb, 6th Main, Tumkur Road, Peenya Post, Bangalore-560058, India.) and Fisher Scientific^®^ (Thermo Fisher Scientific India Pvt. Ltd., 403-404, Delphi, B Wing, Powai, Hiranandani, Business Park, Maharashtra-400076, India.), and used as received without further purification. Unless stated otherwise, the reactions were conducted in oven-dried glassware and under normal atmospheric conditions.

#### 4.1.2. Instrumentation

^1^H-NMR and ^13^C-NMR spectra were recorded on Bruker 500 MHz (Billerica, Massachusetts, USA) spectrometer operating with the ^13^C resonance frequency of 125 MHz and the proton resonance frequency of 500 MHz or Bruker 400 MHz spectrometer (Billerica, Massachusetts, USA) operating with the ^13^C resonance frequency of 100 MHz and the proton resonance frequency of 400 MHz. DMSO-d6 or CDCl_3_ with TMS as an internal standard was used as an NMR solvent. Data from the ^1^H-NMR spectroscopy are reported as chemical shift (δ ppm) with the corresponding integration values. Coupling constants (*J*) are reported in Hertz (Hz). Standard abbreviations indicating multiplicity were used as follows: s (singlet), br (broad), d (doublet), t (triplet), q (quartet) and m (multiplet). Data from ^13^C-NMR spectra are reported in terms of chemical shift (δ ppm). IR spectra were recorded in Thermo Scientific Nicolet Nexus 470 FT-IR spectrometer (6511 Bunker Lake Blvd., Ramsey, MN 55303, USA.) and band positions are reported in reciprocal centimeters. Samples were made as pellet with KBr and recorded. High-resolution mass spectra were recorded on Electrospray Ionization mode on Agilent 6520 (Q-TOF) (Santa Clara, CA, USA) mass spectrometer in positive (ESI^+^) ion mode. Mass spectra were recorded on Perkin Elmer Clarus 600/Shimadzu QP2020 GC-MS spectrometer (Kyoto, Japan) in EI mode. Melting points were recorded with REMI DDMS 2545 (Cama Industrial Estate, Goregaon (East), Mumbai, India). The instrument is calibrated with benzoic acid before the measurement.

#### 4.1.3. Synthesis Methods

The mono-, di- and triconjugated coumaperine derivatives were synthesized as described in detail in the Appendix A. In short, coumaperine derivatives with one alkene bond between the aromatic ring and piperidine are termed monoconjugated coumaperine derivatives and those with two and three alkene bonds are termed di- and triconjugated coumaperine derivatives respectively (Figure 9).

The monoconjugated coumaperine derivatives were synthesized in two steps [28,29]. Knoevenagel condensation of aldehyde and malonic acid in presence of base yields cinnamic acid derivatives. It was then converted to the corresponding acid chloride and subsequently reacted with piperidine allowing us to obtain monoconjugated coumaperine derivatives (refer to ESI for detailed synthetic procedure). CP-215 was demethylated to obtain dihydroxy derivative, CP-237. Demethylation was achieved with AlCl_3_ at room temperature (Figure 10).

Di- and triconjugated derivatives were prepared from the corresponding aldehyde and crotonic/sorbic acid according to our synthetic procedure reported earlier [28,29]. Crotonic/sorbic acid was converted to acid chloride by reacting with thionyl chloride and subsequently reacted with piperidine to obtain crotonyl/sorbyl piperidine. It was then condensed with aromatic aldehydes in presence of base (KOH/t-BuOK) at room temperature to obtain the final product, di/triconjugated coumaperine derivatives (Figure 3)

### 4.2. Biological Assays

#### 4.2.1. Quorum Sensing Inhibition Test with Bio-Reporter *Chromobacterium violaceum* (CV026) and *Agrobacterium tumefaciens* (KYC55)

Thin Layer chromatography (TLC) preparation: All derivatives were dissolved in 100% acetonitrile to a concentration of 40 mM. Then placed 20 μL on the same spot on the TLC (Merk, Darmstadt, Germany) and let the solvent evaporate before use. CV026: few colonies were transferred to liquid LB (Difco Lenox medium, Le Pont se Claix, France) for overnight incubation at 30 °C in a 45 degree in orbital shaker (Gerhardt, Königswinter, Germany). At the O.D. 0.8 3 mL of the starter was added to preheated soft LB agar containing 500 nM of 3-oxo-C6 (K3007, Sigma-Aldrich, Saint louis, MO, USA). The mixture was then loaded on the TLC in a glass plate for O.N. incubation in 30 °C. The next day, the inhibition zone was measured, while QS was taking place that produced a bacterial purple pigment, and when QS is inhibited, no pigment is produced. KYC55 (pJZ372) (pJZ384) (pJZ410): 20 µL from the frozen stock into 25 mL AT media (AT buffer ×20:1.57 M KH_2_PO_4_, pH = 7.3. AT salts mix ×20:300 mM (NH_4_)_2_SO_4_, 13 mM MgSO_4_, 1.4 mM CaCl_2_, 0.4 mM FeSO_4_7H_2_O and 0.26 mM MnSO_4_H_2_O. AT media:50 mL × 20 AT-buffer, 50 mL × 20 AT salts, 10 mL 50% (*w*/*v*) glucose and add 890 mL to a final volume of 1 L). For O.N. at 30 °C in an orbital shaker, into preheated water agar with 25 mL AT buffer and 60 μg/mL Xgal and 5-nM 3-oxo-c8 (O1764, Sigma-Aldrich, Saint louis, MO, USA), 25 mL of the starter was added. Then, that mixture was loaded on the TLC in a glass plate for O.N. incubation in 30 °C. The next day, the inhibition zone was measured. The bacteria produce blue pigment during QS and when QS is inhibited no pigment is produced. KYC55 (pJZ372) (pJZ384) (pJZ410) preservation in tetracycline (1 μg/mL), spectinomycin (100 μg/mL) and gentamicin (100 μg/mL) (all antibiotics from Sigma Aldrich, Saint louis, MO, USA) in AT media O.N. then centrifuged and suspend in 15% glycerol frozen and maintained at −80 °C [31,36].

#### 4.2.2. Diffusion Discs Test

All derivatives were dissolved in acetonitrile to a final concentration of 40 mM. Twenty microliters were loaded onto Whatman filter paper discs and dried. *Bacillus subtilis, Staphylococcus aureus* and *Acinetobacter baumannii* were grown in LB (Lenox) O.N. One hundred microliters. from the starter culture was plated on LB agar (Difco Luria-Bertani medium, Le Pont se Claix, France) dishes. *Streptococcus sobrinus* and *Streptococcus mutans* grown in Brain Heary Infusion (BHI) (Himedia, MS, India) O.N. were similarly plated on BHI agar dishes. Discs were laid and incubated O.N. in 37 °C and the diameter of inhibition of bacterial growth was measured.

#### 4.2.3. NF-κB-Luciferase Reporter Gene Assay

L428 cells stable transfectants with the luciferase NF-κB-Luc reporter gene were generated, as described in Ozer et al. 2009 [35] and were maintained in 500 µg/mL of the G418 (G418 disulfate salt, Sigma-Aldrich, Saint louis, MO, USA). The cells (5 × 10^5^/well in triplicate) were incubated for 2 h in 1 mL of medium containing the solvent (DMSO) or different concentrations of the molecules tested. Cells were then harvested, lysed and monitored by a luciferase reporter assay kit (Promega), according to the manufacturer’s instructions. Measurements were carried out using a luminometer (Promega, GLOMAXTM 20/20 Luminometer, Madison, WI, USA). Data were normalized to the protein concentration in each lysate as measured by the Bradford method (Bio-Rad) and normalized to the control. Experiments were repeated at least three times. Initially, we tested all derivatives that showed QSI activity at two concentrations: 80 and 160 µM (three independent experiments) for NF-κB inhibition activity (data not shown), the dose response of the active derivatives is shown. Cells maintenance was in RPMI medium supplemented 10% FBS, 1% Glutamine and Pen-Strep (all from Biological Industries, Bait-Hemed, Israel).

#### 4.2.4. Quantitation of Activated Nuclear p65 Fluorescence

The A549 adherent human lung epithelial cells were grown in DMEM medium with 10% FBS, 1% Glutamine and Pen-Strep (all from Biological Industries, Bait-Hemed, Israel) and carried by trypsinization (Trypsin EDTA solution B, Biological Industries, Bait-Hemed, Israel). In contrast to L428 cells, IκB is present in A549 cells. We quantified the activated p65 NF-κB in the nucleus of the A549 cells as follows: 25,000 cells were plated per well of a 96-well plate (u-clear F-bottom, Greiner Bio-One) and allowed an overnight attachment at 37 °C, 5% CO_2_. The derivatives (CP-154, CP-158, CP-215, CP-286 and Curcumin (CU) as positive control) were then added at the concentration of 160 µM for 2 h. One-hundred sixty micrometers was chosen as the concentration that inhibited NF-κB in L428 cells by the luciferase reporter gene assay. After 2 h, in order to fully activate NF-κB, TNFα (2.5 ng/mL Recombinant Human TNF-α Protein, R&D system) was added for an additional 15 min. Then, the cells were washed twice with 3% FBS in Phosphate Buffered saline (PBS) before being fixed with 4% paraformaldehyde (16% solution, EM Grade, Electron Microscopy Sciences) in PBS at room temperature for 20 min. Cells were then washed twice in PBS, then twice in 3% FBS in PBS before permeabilization with 0.1% Triton ×100 in PBS for 60 min at room temperature. Cells were then washed twice with 3% FBS in PBS before staining with NF-κB subunit P65 (mouse anti-p65 (F-6) SC-8008, Santa Cruz, Dallas, Texas, USA) diluted 1:50 in 3% FBS and 0.1% Triton ×100 in PBS overnight. The cells were then stained with a secondary antibody AF488 (green) (Goat anti-Mouse IgG (H + L) Alexa Fluor 488 (A-11029]} Invitrogen, Thermo Fisher Scientific, Waltham, MA, USA) diluted 1:400 in 3% FBS and 0.1% Triton ×100 in PBS overnight at RT. Nuclear staining with DAPI (SouthernBiotech, DAPI Fluoromount-G, blue-nuclear. Birmingham, AL, USA) was then added for 30 min, followed by 3× washing with PBS. Cell fluorescence was imaged and quantitated by the Operetta High-Content Imaging System (Perkin Elmer, Waltham, MA, USA) at 40× magnification. Analysis of the pictures was done through the Columbus server of the company where parameters can be defined, such as cell area, nucleus and cytoplasm separately. Different cell populations can be distinguished, and scoring was done on hundreds of cells in each treatment. As an example, see Figure 6. Quantitative fluorescence intensity was averaged, and a relative fluorescence value was given.

#### 4.2.5. Cell Viability by XTT Assay

The human Hodgkin’s Lymphoma, a derived L428 cell line was used as a model. The cells were grown in RPMI Medium 1640 with 10% FBS, 1% Glutamine and pen-strep (all from Biological Industries, Bait-Hemed, Israel). L428 cells do not express the NF-kB inhibitor of NF-kB (IkB) [37]. As a result, NF-κB is constitutively active and expressed mainly in the nucleus. 30,000 cells/well were placed in 96-well plates with the coumaperine derivatives at different concentrations or with vehicle (DMSO) to a final volume of 200 µL. The plates were incubated for 48 h at 37 °C and 5% CO_2_, followed by the addition of 50 µL of the XTT reagent tetrazolium-formazan, (Biological Industries, Bait-Hemed, Israel) was added for 2 h at 37 °C, 5% CO_2_. The mitochondria in live cells oxidizes the tetrazolium-formazan salt generating a color absorbing product at 450 nm. Absorbance measured at 450 nm and 650 nm in an ELISA plate reader (Multiskan Spectrum, Version 1.2. ThermoFisher scientific, Waltham, MA, USA) were recorded. Absorption values at 650 nm were subtracted from the blank and the 450 nm values. The results were normalized to the control (treated with DMSO). To determine the LC_50_ values of the derivatives, the Probit transformation was done only on samples with R^2^ values > 0.85 and only on derivatives that showed significant cytotoxicity.

#### 4.2.6. Statistical Analysis

Two-way ANOVA followed by Tukey’s multiple comparisons test or one-way ANOVA was performed using GraphPad Prism version 8.0.1 for Windows (GraphPad Software, San Diego, California USA, www.graphpad.com).

## Figures and Tables

**Figure 1 molecules-26-02293-f001:**
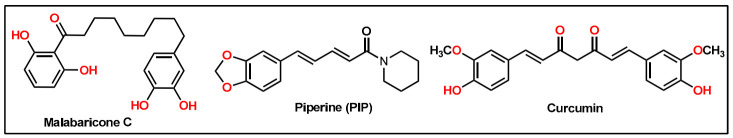
Few examples of natural product-based quorum sensing inhibitors.

**Figure 2 molecules-26-02293-f002:**
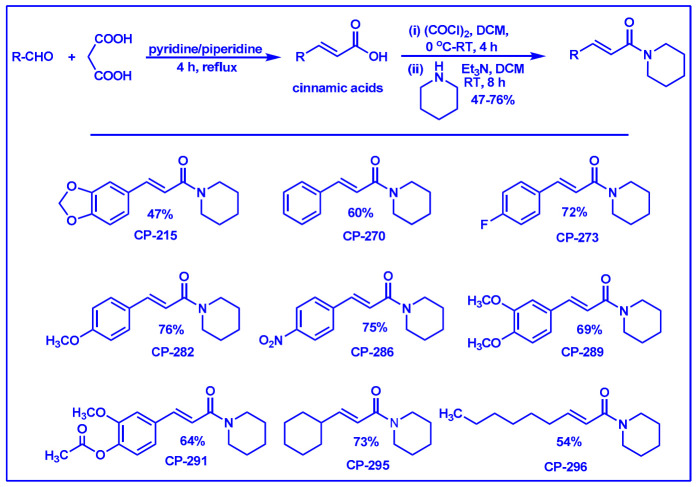
Synthesis of monoconjugated coumaperine derivatives.

**Figure 3 molecules-26-02293-f003:**
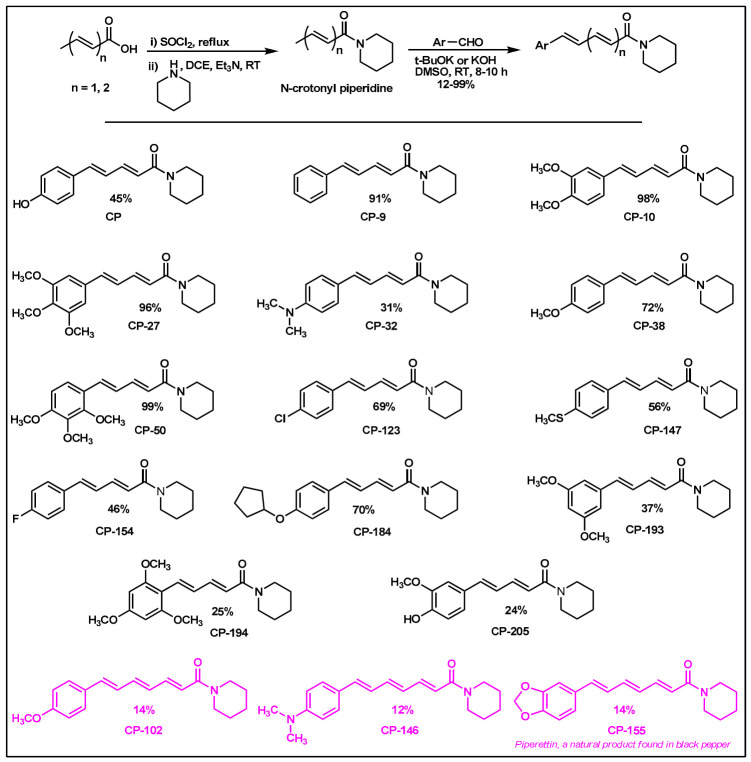
Di and triconjugated coumaperine derivatives synthesized.

**Figure 4 molecules-26-02293-f004:**
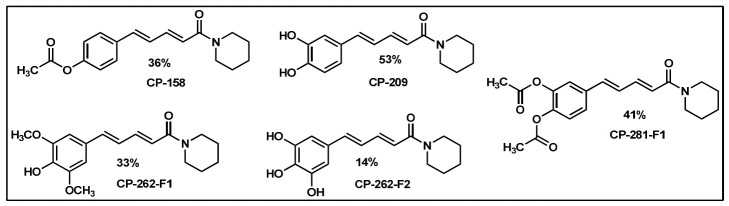
Diconjugated coumaperine derivatives synthesized.

**Figure 5 molecules-26-02293-f005:**
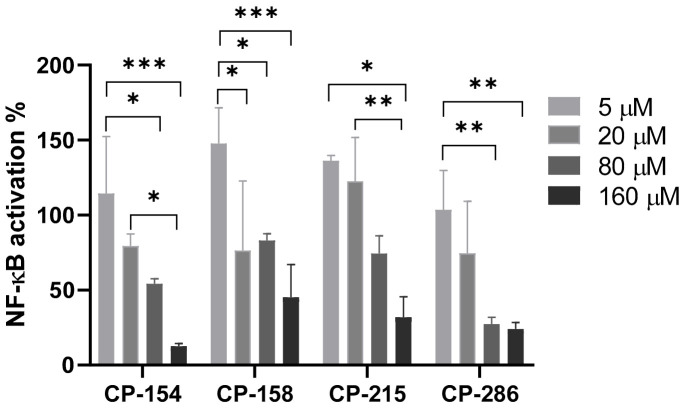
Dose response of Nuclear Factor kappa B (NF-κB) activation using the NF-κB reporter gene luciferase assay. L428 cells were stably transfected with the NF-κB luciferase reporter gene. The cells were incubated with the compounds at different concentrations for 2 h. Only compounds that showed NF-κB inhibition are shown here. The results represent the percentage of NF-kB activation as compared to vehicle (DMSO)-treated cells. All samples were normalized to the protein concentration. Mean + SD, Two-way ANOVA and Tukey’s multiple comparison test. 95% confidence interval (*p*-values * ≤ 0.0332, ** ≤ 0.0021 and *** ≤ 0.0002).

**Figure 6 molecules-26-02293-f006:**
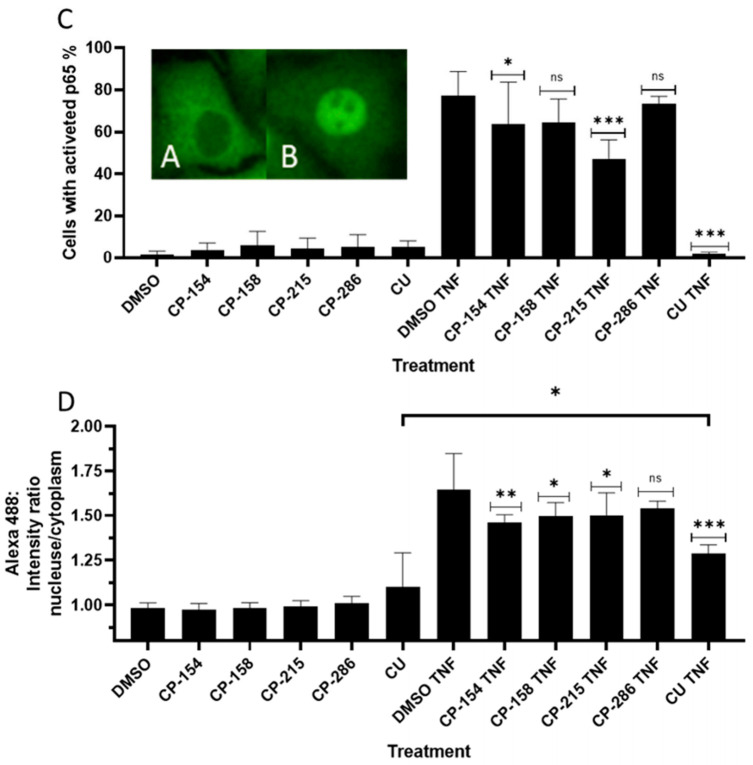
NF-κB inhibition in A549 cells. A549 cells were fixed in paraformaldehyde and immunostained with mouse-anti p65 and fluorescent goat anti-mouse IgG. Localization of p65 (**A**) without treatment- inactive p65 in the cytoplasm. (**B**) After activation with 2.5 ng/mL TNFα for 15 min, activated p65 in the nucleus. All derivatives were added to the cells at 160 µM for 120 min in DMSO (0.16%) then TNFα at 2.5 ng/mL was added for 15 min. (**C**) Activated cells (strong nuclear green fluorescence) as a percentage of all the cells in the analyzed field. (**D**) Ratio of the mean nuclear/cytoplasm fluorescence intensity values in the analyzed fields. Curcumin (CU). Mean ± SD of triplicate samples in two independent experiments. One-way ANOVA comparison of the treatment groups to the control groups DMSO TNF. 95% confidence interval (*p*-values * ≤ 0.0332, ** ≤ 0.0021, *** ≤ 0.0002). For the comparison between the CU and the CU TNF groups- unpaired two tailed t-test with 95% confidence interval, *p*-value is 0.0419 (*).

**Figure 7 molecules-26-02293-f007:**
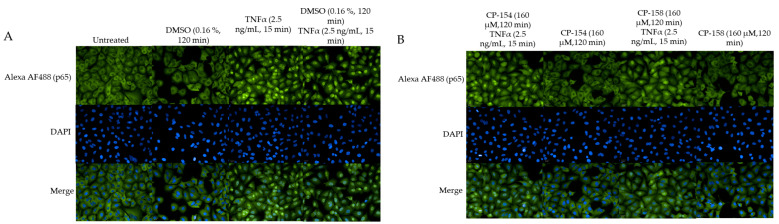
Florescent images of NF-κB inhibition. A549 cells were treated with TNFα (2.5 ng/mL) and/or different positive QS CP derivatives then fixed and immunostained with anti-p65 and anti-mouse IgG conjugate to Alexa AF488 (green) and nuclear staining with DAPI (blue). (**A**) The negative controls were: untreated, treated with the solvent DMSO (0.16%) for 120 min, TNFα (2.5 ng/mL) for 15 min, DMSO (0.16 %) for 120 min with TNFα (2.5 ng/mL) for 15 min. (**B**) CP-154 and CP-158 at 160 μM for 120 min with and without TNFα (2.5 ng/mL) for 15 min. (**C**) CP-286 and CP-215 at 160 μM for 120 min with and without TNFα (2.5 ng/mL) for 15 min. (**D**). Curcumin (CU) was used as a positive control with the same experimental design as in (**B**) or (**C**).

**Figure 8 molecules-26-02293-f008:**
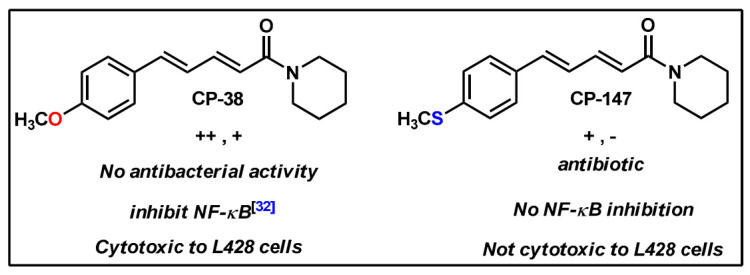
Structure activity relationship of methoxy (**CP-38**) and thiomethyl (**CP-147**)-diconjugated coumaperine derivatives.

**Figure 9 molecules-26-02293-f009:**
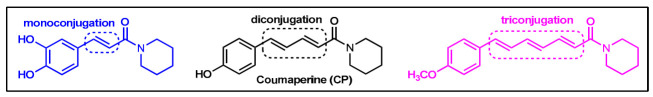
Mono-, di- and triconjugated coumaperine derivatives.

**Figure 10 molecules-26-02293-f010:**
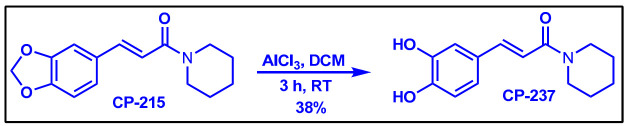
Synthesis of monoconjugated coumaperine derivatives, CP-237.

**Table 1 molecules-26-02293-t001:** Degree of Quorum Sensing Inhibition (QSI) coumaperine derivatives *.

**Entry**	**Monoconjugated-CP**	**Degree of QSI of CV026**	**Degree of QSI of KYC55**
1.	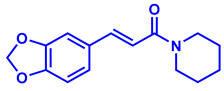 CP-215	+++	+++
2.	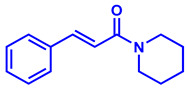 CP-270	+++	+++
3.	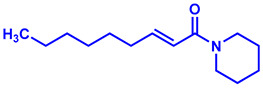 CP-296	+++	+++
4.	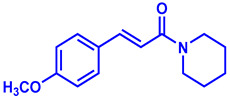 CP-282	+++	++
5.	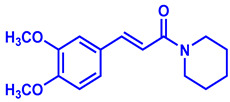 CP-289	+	+++
6.	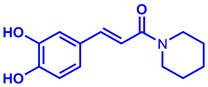 CP-237	++	++
7.	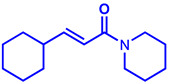 CP-295	++	++
8.	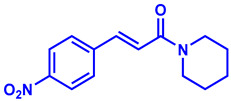 CP-286	+	++
**Entry**	**Diconjugated-CP**	**Degree of QSI of CV026**	**Degree of QSI of KYC55**
9.	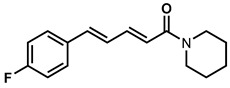 CP-154	+++	+++
10.	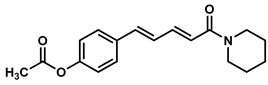 CP-158	+++	++
11.	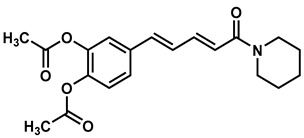 CP-281 F1	++	+
12.	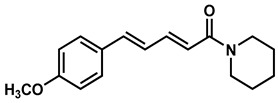 CP-38	++	+
13.	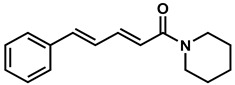 CP-9	++	+
**Entry**	**Triconjugated-CP**	**Degree of QSI of CV026**	**Degree of QSI of KYC55**
14.	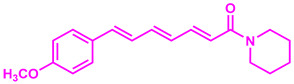 CP-102	++	+

* Two tests were carried out using the reporter bacteria: Agrobacterium tumefaciens (KYC55) and Chromobacterium violaceum (CV026). Bacteria were grown overnight in soft agar with the appropriate autoinducer on a Thin Layer chromatography (TLC) plate where the different compounds were dried on 40 mM in ~20-µL acetonitrile, and their degree of QSI was measured semi-quantitatively by color intensity (+ low inhibition, ++ medium inhibition and +++ strong inhibition). Experiments were repeated three times, and the results were determined by considering at least 2 out of 3 experiments with the same results.

**Table 2 molecules-26-02293-t002:** Antibacterial activity of coumaperine derivatives *.

Compound Code	*Bacillus subtilis* (mm)	*Staphylococcus aureus* (mm)	*Streptococcus sobrinus* (mm)	*Streptococcus mutans* (mm)	PA01 (mm)	*Acinetobacter baumannii* (mm)
CP-147	-	9	-	-	-	-
CP-154	7	11	9	-	-	-
CP-9	8	9	9	-	-	-

* Discs with 40 mM of compound in 20 µL solvent (DMSO) were placed on petri dishes with different bacteria. The numbers represent the diameter of growth inhibition (mm). (-), no activity.

**Table 3 molecules-26-02293-t003:** Summary of Biological activities *.

Compound Code	QSI	Antibacterial	NF-κB Inhibition	Cytotoxic
CP-9	+	+	+	+
CP-154	+	+	+	+
CP-158	+	−	+	+
CP-215	+	−	+	−
CP-286	+	−	+	−

* Five compounds with QSI and NF-κB inhibitory properties were tested for antibacterial and cytotoxic activities. (+) active (−) inactive.

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
