# Peer review of "Quorum Sensing and NF-κB Inhibition of Synthetic Coumaperine Derivatives from Piper nigrum"

_molecules, 2021, doi:10.3390/molecules26082293_

Round 1

Reviewer 1 Report

Kadosh et al. describe the synthesis of Coumaperine derivatives and their characterization in regard to quorum sensing, antibacterial, “anti-inflammatory” activities, and cytotoxicity (in L428 cells), demonstrating overlapping but also distinct activities for the different compounds in these assays.

This is essentially a nice story and interesting approach; However, I have one major critique, namely that the authors have NOT investigated the anti-inflammatory activity, but NF-kB activation. Although NF-kB can be considered a major driver of this response, this cannot be termed the same (likewise, NF-kB inhibition cannot be set equal to e.g., immunosuppression though it is involved in T cell activation, etc…). Also, especially L428 (due to constitutive active NF-kB), but also A549, although valuable for NF-kB analysis, are not the favorite models for inflammation. Therefore, the authors should use established models such as macrophage or endothelial cells /lines and assay for the a few relevant inflammatory genes on the mRNA and protein level. Otherwise, the term anti-inflammatory activity should be deleted from the entire manuscript and especially the title, and replaced by something like NF-kB inhibitory effect (which is by itself already an important aspect).

Minor comments:

  1. M&M 4.2.4.: A549 cells are epithelial, not endothelial!
  2. Some wording should be evaluated for proper expression, e.g. Intro middle 1st para: “Herein, two examples ..”: this is spoken language, not written, and others.

Reviewer 2 Report

  1. Some words were combined, for example, Myristicacinnamomea, please revise.
  2. Indicate next to each section that the raw data are shown in the supplementary file 
  3. Monoconjugated coumaperine derivatives were synthesized in 47-76% yields, please clarify, what does it mean in 47-76% yields? and throughout the text with the others.
  4. NF-kB please change k to κ throughout the text.
  5. Figure 5, the compounds showed dose dependent inhibition, have you calculated the IC50? also, comaprison with a standard compound is recomended to evaluate the activity of the compounds.

Round 2

Reviewer 1 Report

The authors have replaced anti-inflammation by NF-kB inhibition (and taken the easy way) but that is fine as already mentioned. Also the text was refinished extensively and substantially improved. Therefore, all my points have been addressed satisfactorily.

Reviewer 2 Report

The authors answered the raised comments